# Collaborative Build of Statistics and Machine Learning Analysis Pipelines for Cosmology and Particle Physics

## Abstract

In a full-time, week-long project after a Statistics and Machine Learning course, students grouped in "Collaboration" of Teams build pipelines to analyse scientific data: to prove the existence of the Higgs boson from the Large Hadron Collider data or Dark Energy from supernova surveys. They have the opportunity to implement together a variety of tools and concepts from physics, data processing, Statistics and Machine Learning, borrowed from the course, earlier training or resources from the Internet.

## 1. Introduction

Quite often, when teaching Statistics and Machine Learning, specific topics are introduced one after the other and exercises (or hands-on tutorials) are self-contained. We have set up in 2019 (and repeated in 2020 and 2021) a broader week-long project where scientific data analysis pipelines are built by the students, complementing a 30 hour introductory course on Statistics and Machine Learning for particle physics and cosmology. The objective is to give them a thorough experience by connecting the various elements they have been taught in order to obtain the best possible measurement. The topics are broad and students must organise themselves in teams and collaborate in the same spirit as international experimental collaborations.

The original idea comes from another department in the University where students are working in teams to design a synchrotron beam line, from the extraction of the X-rays through various collimators onto a target. Here, data measured by a scientific instrument (an experiment at the Large Hadron Collider or large telescopes) are processed step by step until a measurement is obtained, including uncertainties.

## 2. Organisation

### 2.1. Overview

The project lasts one week full time, starting with a short kick-off meeting Monday morning, and ending with an af-

ternoon of presentations on Friday afternoon. Prior to the project week, students are asked to group themselves into independent "collaborations" of about 25 students. The topic of each collaboration is then randomly assigned. We are three tutors to mentor four collaborations, circa 100 students (two tutors would be barely possible).

The 30-minute kick-off meeting on Monday allows tutors to give a very brief introduction to the projects and what each team is expected to do. Students are encouraged to find the necessary information in their courses and on the Internet.

The collaborations then have until noon to elect a "spokesperson" who will coordinate the different teams. Before 5 pm, the spokesperson must send a face book with the members of each team of 5 students.

A large open space is reserved for the project for the entire week[1], though the students are free to settle anywhere, outside scheduled meetings. Tutors remain available there all the time for scheduled or impromptu discussions.

Communication between tutors and students either occurs directly or through the official University chat/meeting tools, where dedicated channels are opened for each collaboration. Students are free to set up their own chat tools where they can exchange without tutor knowledge, to ease exchanges between them and to encourage the collaborative spirit. Each team as its own tasks but they are interdependent through the global collaboration results. The interfaces between teams are not fully spelled out at the beginning and require some negociation, so that teams have to share data, algorithms, concepts and tools along the week.

The spokespersons send every evening a status report, a bullet list of one page describing the status of each team. The status report is discussed every morning with the spokesperson in the presence of the whole collaboration so that issues are identified. If a team is stuck, or heading in a clearly wrong direction, a tutor will talk with them to help them overcome the hurdle. Initiatives from the teams are favoured, the mentoring from the tutor should be as light as possible.

---

[1]Due to the Covid-19 pandemics, in 2020 the project was entirely conducted online, while in 2021, 10% of the students were online, the rest being onsite.

A second checkpoint takes place in the early afternoon with the spokesperson alone, often deeper on a specific topic or regarding potential problems in managing collaborations.

Friday morning is devoted to the preparation of a 45-minute presentation (+15 minutes for questions) by each collaboration. The presentation is introduced by the spokesperson, followed by a speaker in each team. The speaker is chosen at random from each team by the tutor and announced at noon, after the collection of deliverables, to ensure that everyone is fully engaged until then. Questions from students are encouraged.

The whole pipeline is implemented in Google Colaboratory Jupyter (Kluyver et al., 2016) Python notebooks, which was introduced and used during the practice sessions of the course. This solution was chosen for two reasons: it does not require any individual configuration and it is scalable. The initial dataset and intermediate steps are shared through Google Drive cloud storages. All students have laptops connected to a robust wireless network. Some students may also have installed an Anaconda environment on their laptops and developed on them.

## 2.2. Student background

The course is designed for undergraduate students in engineering schools, at the university Bachelor level. After a general training, with majors in mathematics and physics, students choose their areas of interest (here physics) during their curriculum. The proposed course is part of the "data analysis" sequence of their curriculum. Most of our students had no background in the topics to be covered, other than a basic understanding of quantum mechanics, and basics of statistics and probability. Students have all received formal training in the Python language, but the level of proficiency vary greatly between students.

The course is designed as a series of introductory lectures to Cosmology and Particle Physics allowing to understand the type of data and the purpose of data analysis. The Statistics and Machine Learning concepts listed in 3.2 are covered. Several practice sessions with exercises are scheduled to get students used to coding and data manipulation.

## 2.3. Deliverables

The main deliverable is the 45-minute presentation. A pdf file of the slides (without animation!) has to be received by the tutors on Friday at noon. The presentation has to be understandable to students in other collaborations (e.g. the cosmology collaborations students should understand the Higgs boson presentation). In addition, notebooks also have to be delivered on Friday at noon. For instance, for the Higgs pipeline, 3 notebooks are requested: one that runs the pipeline from start to finish, except for the training, one that

trains the BDT (Boosted Decision Trees), one that trains the NN (Neural Networks). Additional notebooks to illustrate specific studies are also welcome. In all cases the notebooks should be clearly commented on what they are doing, all graphs should be clearly labelled.

To track the progress of the project, the spokesperson sends a single page progress report by midnight each day. In addition, each student must fill out an online form by midnight with a few sentences about what they did, who they worked with, any difficulties and the plan for the next day. They are also encouraged to express their satisfaction/dissatisfaction.

## 2.4. Evaluation

The evaluation stage of students' progress, work, and involvement during the week is an important and necessary element of the teaching experience. As indicated earlier, specific indicators for monitoring the on-going work were developed. The primary one is a comprehensive daily report with a short 5-10 minute presentation by the spokesperson for each collaboration. It is the primary source of evaluation of each team, complemented by ad-hoc discussions with the teams on the initiative of the teams themselves, the spokesperson or the tutors. Tutors remain watchful to the flexibility of each team to adapt to upstream and downstream teams requests. Most often, the difficulties are either technical or relational.

In addition to these overall reports, individual reports are requested through online forms to track each student's work and consistency with the team's messages. These individual reports help to gauge individual efforts, as "invisible" students may contribute significantly to the team effort through private channels. The tutors may schedule face-to-face discussion with the few students who seems less engaged. Overall, the careful evaluation of teams and students is the main reason to limit the attendance to 50 students per tutor.

The final evaluation is based on a 45-minute presentation by each collaboration of the work done throughout the week. The spokesperson is responsible for the introduction and overall consistency of the collaboration's message, and the overall conclusion. Each student's final grade is evaluated primarily on the team performance, the quality and diversity of the studies performed, technical and interpersonal skills, and the Python notebooks quality. Tutors can provide additional bonus to spokespersons for the extra workload and for individuals who provided outstanding efforts for the overall collaborations.

# 3. Academic content

## 3.1. Brief description of the pipelines

Two alternative projects are proposed, which have been devised to credibly reproduce high-profile, Nobel prize level scientific achievements. Despite the sequential nature of the pipelines, all teams can get started immediately, before connecting to each other.

### 3.1.1. THE HIGGS PIPELINE

Students are presented a special version of the Higgs Machine Learning (HiggsML) challenge dataset (ATLAS collaboration (2014), 2014) and associated documentation. This dataset was created for a 2014 Kaggle challenge to investigate Machine Learning algorithms on a high-profile High Energy physics task: extracting the Higgs boson signal from overwhelming background noise. The dataset is a csv file containing tabular data with 17 "primary" features corresponding to the measured parameters of the particles from the simulated proton collision. This is a classification task with a specific figure of merit, the Approximate Median Significance (AMS) that evaluates discovery potential. Prior to the project, students have studied notebooks for a BDT or NN trained on a similar dataset, which they can adapt to this new dataset as a starting point.

The pipeline to be built has five components to be addressed by one team each.

- **Feature Engineering (FE)**: the original HiggsML dataset has been stripped of all "derived" features, computed using the knowledge of physics experts. The FE team should first rebuild the derived features (given by mathematical formulas), study their importance with the BDT and NN classifiers downstream, and propose new features.

- **Boosted Decision Tree (BDT)**: the BDT team should train a BDT (actually two: XGboost and LightGBM) to maximise the AMS, first on the original dataset with only primary features, then with the additional features provided by the FE team. They should proceed with hyperparameter optimisation (HPO) and other studies as listed in section 3.2.

- **Neural Network (NN)**: the NN team should train a Neural Network to maximise the AMS, first on the original dataset with only primary features, then with the additional features provided by the FE team. They should proceed with HPO (in particular optimise the architecture of the NN) and other studies as listed in section 3.2. Given the few minutes of training time compared to few seconds for the BDT, they should organise well to optimally cover the HPO space.

- **STAT**: the STATISTICS team has to develop a likelihood framework based on the output of the two previously trained model (BDT and NN) in order to incorporate shape discrimination between signal and background and to exploit the modelling power of the algorithms to increase the statistical significance of signal detection.

- **SYST**: the SYSTEMATICS team should become familiar with the entire data analysis pipeline and develop a framework to re-evaluate the trained models (BDT and NN) under different conditions when the nominal working assumptions are wrong or biased by some amount (e.g. a +3% error on background estimate,...). A script allowing to alter the original dataset is provided to them. Thus, they should evaluate the impact of the different biases on the input dataset to investigate systematic effects and, if possible, find ways to provide results robust to these effects during model (BDT and NN) training stages.

Ideally, they should iterate to provide the best performance on the statistical significance of the Higgs signal determination and have the lowest dependence on the systematics which have natively a stronger impact.

### 3.1.2. THE COSMOLOGY PIPELINE

Students are presented with two simulated datasets: Type Ia supernovae and Cosmic Microwave Background (CMB) data. They were both simulated assuming cosmological parameters (Hubble-Lemaître constant $H_0$, matter density $\Omega_m$ and Dark Energy density $\Omega_\Lambda$) distinct from those measured for our Universe. The aim of the project is to build a pipeline for each probe allowing to go from raw observations to constraints on these parameters as well as joint analysis breaking degeneracies of each probe. The different Work-Packages are organised in the following manner:

- **WP-SN1: Supernovae detection from a series of images:** The input data are a series of images containing stars and galaxies with known magnitude and location. The images are noisy and taken under different conditions. Supernovae are detected on image differences with respect to a reference one with no supernovae. Deliverables are a list of SNIa candidates for each field.

- **WP-SN2: Photometry of detected supernovae from a series of images:** The input data is similar to that of WP-SN1 but with a list of SNIa candidates coordinates. The deliverable is a measurement of each calibrated SN's flux and error bars on image differences.

- **WP-SN3: Supernovae lightcurve fitting and cosmological constraints:** The input data is a set of calibrated lightcurves for a number of SNe. The objective

is to build a Hubble diagram from these SNe measuring their brightness at maximum including various corrections. A Markov-Chain-Monte-Carlo (MCMC) approach provides cosmological constraints on ($H_0$, $\Omega_m$, $\Omega_\Lambda$) from the Hubble diagram.

- **WP-CMB1: From Time-Ordered-Data (TOD) to Cosmic Microwave Background (CMB) Maps:** The input data are noisy TOD along with the corresponding pointing. The deliverable is a projected map with uncertainties of these TODs maximising signal-to-noise ratio through time-domain filtering.

- **WP-CMB2: From CMB Maps to CMB angular power spectra:** Input data is a simulated observed map of the CMB including inhomogeneous noise and the corresponding coverage map. The objective and deliverable is to calculate the CMB angular power spectrum from this map, uncertainties are determined through a Monte-Carlo simulation.

- **WP-CMB3: Cosmological constraints from CMB angular power spectra:** The input data is an angular power spectrum of the CMB with error bars. The objective is to constrain cosmological parameters using a MCMC approach and theoretical power spectra.

Finally, WP-SN3 and WP-CMB3 are expected to perform a joint MCMC analysis of their dataset in order to obtain the final measurements on the cosmological parameters.

### 3.2. Statistics and ML tools and concepts to be implemented

During the construction of the pipelines, students have the opportunity to implement many concepts and use many tools. Although the ones from the course are sufficient to obtain reasonable results, they are encouraged to look for more. These tools and concepts are listed below without any details.

In physics: special relativity, and how the Higgs boson was discovered! General relativity, to prove the existence of dark energy!

In statistics and data processing : data cleanup (check those NaNs!), signal processing and filtering, Markov Chain Monte Carlo, scientific plots with labels and legends. Students have to practice maximum likelihood and least squares estimators, check the consistency of the results and quantify the associated uncertainties, either by the classical definition of confidence intervals or by Monte Carlo simulation.

In Machine Learning: use of SciPy and Scikit-Learn (Pedregosa et al., 2011), feature engineering (Machine Learning does not work miracles), feature normalisation, feature importance with permutation importance, feature selection,

Boosted Decision Trees with XGBoost (Chen & Guestrin, 2016) and LightGBM (Ke et al., 2017), Neural Networks with Keras (Chollet et al., 2015), importance sampling, train vs. test splitting, cross-validation, overtraining, model serialisation, classifier evaluation, ROC curve, significance curve, hyperparameter optimisation (manual, grid search, random search), clustering (DBScan (Ester et al., 1996)).

In addition, students soft skills are being exercised as well. Team work of course, at the level of the team and at the level of the collaboration, scientific discussion, presentation of results in a compelling way.

## 4. Outlook

By the end of the week, students manage to run a full data analysis pipeline, the details of which they expose in a way that shows (in most cases) they really understand what they are doing. For the Higgs pipeline, we had in mind that the students would iterate their model in order to minimise the overall uncertainty. In practice no collaboration have had the time to do it so far. If we would provide a functional minimal pipeline with clear interfaces to get started, they would spend less time with technicalities but they probably would learn less overall.

Also, using Git is certainly a better way to exchange code than Google Colab, but few students master contributing to a Git repository (as opposed to downloading from it), so that using Git would bar a large fraction of the students to contribute. On the other hand, the topics are rich enough so that the job of any team is never complete; good students can always do more in-depth studies[2]. It is up to the tutors to adjust the balance between autonomy (at the risk of achieving little) and strong supervision (at the risk of falling back to more usual exercises).

This type of project is very different from a challenge "*à la Kaggle*" where a single figure of merit is optimised. What matters here is that students overcome the various difficulties with minimal guidance and are able to perform a number of small studies on their own. We have often seen teams perform unexpected studies with quite interesting results. This type of project can certainly be adapted to other datasets from different domains, and with students with different level of expertise.

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
