# OpenReview forum: "Collaborative Build of Statistics and Machine Learning Analysis Pipelines for Cosmology and Particle Physics"
_ecmlpkdd.org/ECMLPKDD/2021/Workshop/TeachML — Submitted to TeachML 2021_

### Official Review · Reviewer_2896 · 2021-07-09
**ML pipeline development to mimic a scientific collaboration**

**Rating:** 5
**Confidence:** 3

**Review:**

This submission describes a one-week intensive course for undergraduate physicists, designed to follow a 30-hour course in Machine Learning and Statistics for particle physics and cosmology. The course is designed to mimic a scientific collaboration such as ATLAS -- students are split into "collaborations" of 25, which are subdivided into teams of 5. The teams each work on a component of the problem pipeline, but must coordinate with other teams and with a spokesperson. The spokesperson provides daily updates, and the teams each give a presentation at the end of the week.

There are many things I like here, some things I dislike, and some things that I did not fully understand. In some cases, the "dislikes" may in fact be "didn't fully understands".

Things I liked:
* I haven't seen a syllabus like this that is designed to mimic a large collaboration. I think it's a really neat idea, and teaches a lot of "soft skills". This is particularly important, given the audience -- many of these students are likely to work in such collaborations in future.
* The randomization of who gives the presentation is a great way to ensure everyone puts in the work behind the presentation.
* Realistic datasets and problems that tie in with the students interests and previous courses.

Things I didn't fully understand:
* I found the description of the Cosmology project confusing. In part, this might be because I have some background in HEP, but none in cosmology... but even accounting for that bias, the Cosmology project seemed much less clear than the HEP project, and there seemed to be little standardization between the two sections of the paper. In particular:
- The rest of the paper describes collaborations of 25, with 5 teams. The HEP section makes clear what the teams are. Here, the teams aren't mentioned, instead the authors talk of "work-packages". I assume these correspond to teams; however there are six work-packages, not 5.
- While the HEP project goes into details about what methods are expected (e.g. the BDT team are expected to implement XGboost and LightGBM), the descriptions for the Cosmology project are vague. What methods are they using? Do they have packages, or are they implementing methods? Have they seen similar examples in the 30 hour course?
* The list of tools and concepts is very long. Have they really learned these in sufficient depth in a single 30 hour course?
* How do students get around the sequential nature of the projects? For example, it seems WP-SN2 requires candidate SNe from WP-SN1, and WP-SN3 requires curves from WP-SN2. Similarly, the STAT team in the HEP task relies on the output of the NN and BDT groups. Are they given sample data to get started on?
* I was a little confused by the STAT task. What do you mean by "exploit the modeling power of the algorithms to increase the statistical significance"? Does the likelihood framework assume that the BDT and the NN output well-calibrated probabilities?


Things I didn't like:
* Large, intensive group projects like this tend to be much better suited to students without external responsibilities. The timelines seem to assume that students will be working together late into the night, which places an undue burden on students with children or other responsibilities.
* The workload of the coordinator seems to be much higher than the other members of the collaboration. It appears the coordinator is also a member of a team? The authors do mention giving some extra credit to coordinators, but this opens up unfairness since that extra credit is only available to one person.
* As mentioned above, I like the collaboration format. But, as stated, the tasks are highly sequential. In an actual collaboration, this isn't a problem as scientific collaborations occur over a long time period, but in a 5 day intensive course, it seems that the workload will have periods of waiting combined with intensive periods of activity. While in some cases, it is possible to implement models before having the data, it doesn't promote good data science practices to not consider the data when designing models.

Things I would have liked to have seen:
* An analysis of the workload and average grade across the different groups -- are the tasks similarly challenging?
* Feedback from students -- what did they like or dislike?
* What are the expected time requirements of the course? Are students taking other courses simultaneously?

---

### Decision · Program_Chairs · 2021-07-23

**Decision:**

Reject

**Comment:**

Thank you for submitting this year to the Teaching ML workshop. The reviewer and PCs agree that this paper is not ready for publication. Specifically, the PCs do agree with reviewer concerns about student workload during workshop. For a future iteration, the PCs would also recommend highlighting only one pipeline in section 3 and mentioning the existence of the remaining pipelines briefly.

We encourage the authors to keep up their efforts in the field and act upon the suggestions made. We would love to see a submission from you in the future. We cordially invite you dial in for the workshop itself to be part of our community and make contributions there. We are looking forward to hearing from you in the future.